# Breast tenderness and swelling experiences related to menstrual cycles and ovulation in healthy premenopausal women: Secondary analysis of the 1-year "Prospective Ovulation Cohort"

Mary Wood[1], Sonia Shirin[2,3], Azita Goshtasebi[2,3,4], Jerilynn C. Prior [2,3,4,5]*

**1** University of British Columbia (UBC) Medicine Undergraduate Program (2024), Vancouver, British Columbia, Canada, **2** UBC Centre for Menstrual Cycle and Ovulation Research (CeMCOR, www.cemcor.ubc.ca), Vancouver, British Columbia, Canada, **3** BC Women's Health Research Institute, Vancouver, British Columbia, Canada, **4** Division of Endocrinology, Department of Medicine, UBC, Vancouver, British Columbia, Canada, **5** UBC School of Population and Public Health, Vancouver, British Columbia, Canada

* jerilynn.prior@ubc.ca

## Abstract

Breast tenderness and swelling are associated with premenstrual symptoms but are not well described in healthy women. In this 1-year prospective observational study, we examined daily breast tenderness and swelling to determine whether differences existed between normally ovulatory and ovulatory disturbed (short luteal phase and anovulatory) cycles in a cohort of community dwelling, non-smoking, healthy pre-menopausal women. Enrolment required two consecutive normal-length and normally ovulatory cycles by Quantitative Basal Temperature© analysis. Women (n = 53) ages 20–41 recorded their daily breast experiences in the Menstrual Cycle Diary© across an average of 13.6 cycles. In all 720 cycles, the median breast tenderness was 1.4 (on a 0–4 scale, range 0.0–3.0), in cycles with a mean length of 28.1 days (95% CI 27.5–28.8). Comparison of breast tenderness and breast size (changes from usual) parameters between all normally ovulatory cycles and all ovulatory disturbed cycles in the whole cohort showed significantly higher levels in normally ovulatory (luteal length ≥10 days) in both Breast Tenderness Score [intensity X duration in days; 6.0 (range 1.0–14.0) vs. 3.0 (0.0–11.0) (P=.005)] and breast size [4.0 (2.0–4.0) vs. 4.0 (0.0–4.0) (P=.034)]. However, within-woman in the forty-seven women with both normally ovulatory and ovulatory disturbed cycles, breast tenderness (intensity, duration, and Breast Tenderness Score), did not differ between normally ovulatory cycles and cycles with ovulatory disturbances. This study also demonstrated that in all ovulatory cycles, the timing of breast tenderness increased in parallel with breast swelling; the maximum for both was in the late luteal phase.

**Data availability statement:** We have uploaded the study data to Borealis, UBC's research data repository. Here is the link: https://borealisdata.ca/privateurl.xhtml?token=c9a-3da20-83f4-4a8b-b3f2-eda631b29a2a Data requests can be made to Dr. Eugene Barsky or the UBC Research Data Team research.data@ubc.ca

**Funding:** The author(s) received no specific funding for this work.

**Competing interests:** The authors have declared that no competing interests exist.

## Introduction

Breast tenderness associated with the menstrual cycle is commonly reported as noted by 68% of a random sample of ~900 premenopausal women [1]; it often co-occurs with breast swelling [2,3]. Breast tenderness is likely 'normal' when women experience it for fewer than five days/cycle prior to flow and if it decreases with the onset of flow [2–4].

Although there is no clear etiology for these cyclical breast changes, several physiological mechanisms are possible. These include increased dietary fat intake' [5]; higher prolactin levels leading to fluid retention in the breast [6]; imbalance in estradiol and progesterone activities on breast proliferation, differentiation, and maturation [7]; and'anovulatory cycles' [8].

However, most menstrual studies do not assess ovulatory status or use non-valid methods [9,10]. Thus, there has yet been no accurate assessment of breast tenderness and swelling experiences related to ovulatory status [9]. Previous studies that *have* confirmed ovulation had durations of only 1–2 cycles [11–13], often used retrospectively reported data [12,14], or included only clinical participants [2,10,14].

Both women and clinicians need to know what is expected or 'normal' for menstrual cycle-related breast swelling and tenderness. We also need to know if breast changes across a cycle reflect ovulatory status. Given the long-term consequences associated with anovulation, including decreased bone and cardiovascular health [7], breast experiences might allow for the identification of women with predictable, month-apart cycles who have ovulatory disturbances (called *Subclinical Ovulatory Disturbances*, SOD). These women may benefit from consideration of cyclic progesterone therapy [15,16].

The primary objective of this study was to determine whether prospectively recorded breast tenderness differed between normally ovulatory and ovulatory-disturbed cycles within-woman in a cohort of healthy, community dwelling women with ~month-apart and luteal phase length- (LL) documented cycles. We hypothesized that there would be an increase in the intensity and duration of breast tenderness in cycles with SOD given their relative or absolute absence of progesterone [7].

## Methods

### Participants

These data were prospectively collected between 1984 and 1986 for a primary study, published in 1990 whose outcome was spinal bone change [16]. To enrol, women needed two consecutive normal-length (21–36 day) [17] and normally ovulatory (LL ≥10 days by Quantitative Basal Temperature© [QBT©]) cycles [16].

This cohort of community dwelling, premenopausal women were between ages 20–41 years, with body mass indexes (BMI) between 18.5–24.9, and were in overall good physical and emotional health [16], non-smoking and had not taken any form of hormonal contraception within 6 months [16].

The 66 women who completed the initial trial became the "Prospective Ovulation Cohort" (POC). Data from those 53 with complete menstrual cycle experience records (the Menstrual Cycle Diary©, MCD, was optional in the original study) plus cycle and ovulation documentation over ≥8 (mean=13) cycles were analyzed.

## Menstrual Cycle Diary© (MCD©) and Quantitative Basal Temperature© (QBT©) analysis

All in the POC were asked to record the start of each cycle. The MCD© [18] recording daily cycle and general experiences, was completed before bed, and included breast tenderness and change from usual in breast size by ordinal scales [18].

Assessments of ovulation and LL used the Quantitative Basal Temperature© (QBT©) analysis of first morning temperatures recorded in the MCD©. QBT© has been twice blindly validated: against the serum luteinizing hormone (LH) peak, and versus a within-cycle three-fold rise in the follicular-to-luteal urinary progesterone metabolite level [19,20]. All thermometers were from a single batch and women recorded to the nearest 0.05°C (Becton Dickinson No. 4009) [16]; they also noted external factors that could have affected their first-morning temperatures.

By QBT©, cycles were categorized as being either normally ovulatory (LL≥10 days), short luteal phase (<10 days) or anovulatory (no significant temperature rise or for <4 days).

## Breast experiences - Breast tenderness and swelling

Using the MCD©, breast tenderness intensity was recorded on a five-point ordinal scale (0 = None, 1 = Minimal, 2 = Moderate, 3 = Moderately Intense, and 4 = Very Intense) [16]. A Breast Tenderness Score was derived by multiplying this intensity by its duration in days. We calculated mean breast tenderness per woman over all cycles and the number of days of an intensity > 0.

Breast size changes were also recorded in relation to 'usual' which was the mid-point on a five-point scale. These data were collected by capital letters with U = usual (3), and two letters above and below. Letters were converted into numbers: 1 = much less, 2 = a little less, 3 = usual, 4 = a little increased, and 5 = much increased [17]. Mean change in breast size from each woman's perceived usual (including days with MCD© data <3 or >3), and days of size change were evaluated across all cycles for the entire cohort and within-woman between normally ovulatory and SOD.

## Statistical analysis

Demographic, anthropomorphic, exercise, menstrual and reproductive health variables for all women were assessed for distribution and described as mean (95% confidence interval) or median, range across two groups split according to the median intensity of breast tenderness.

Based on data distribution (normally vs. non-normally distributed), statistical analysis used either ANOVA or a Mann-Whitney U tests to identify differences between breast tenderness groups.

Mean breast tenderness and changes in breast size were compared between all normally ovulatory cycles (LL≥ 10) and SOD cycles with QBT© data in the 53 women using a Mann-Whitney U test.

Forty-seven cohort women experienced *both* normally ovulatory and SOD cycles during ~1-year. We used Wilcoxon Signed Rank Test to compare median breast tenderness intensity, duration and Breast Tenderness Score, plus breast size changes and days of change from usual, in a within-woman comparison of normally ovulatory versus SOD cycles.

A multiple linear regression assessed demographic, anthropomorphic, exercise, menstrual and reproductive health variables for all 53 women related to continuous documentation of breast tenderness. In addition, breast tenderness and size changes were both described graphically in all ovulatory cycles oriented to the QBT© day of ovulation. Breast tenderness and size correlations were examined by Pearson correlation coefficient. Statistical analyses were performed with SPSS (version 29; Armonk, NY: IBM Corp) and R version 4.3.0.v with probabilities of <.05 considered likely important.

## Ethics approval

The original study, with data collected approximately between 1984 and 1986, was approved by the Research Office of the University of British Columbia in 1984 (#C84-007). An overarching protocol and ethics proposal to analyze previously collected, unanalyzed data was subsequently approved for the 'Menstrual Cycle and Ovulation-Related Experiences in Healthy Premenopausal Women: the 1-year Prospective Ovulation Cohort (POC)' (#H21-03130) [21] as well as for this

discrete breast-focussed project of the POC study (#H22-00695). The de-identified data were released for statistical analysis on May 5, 2022. Participants were not financially compensated for study participation but learned their own results and the whole study's findings before publication [16].

### Participant consent

All study participants were community dwelling women, not patients, and provided written informed consent to the original study protocol.

## Results

Women participants averaged age 34 years (95% CI 32.4, 35.4), BMI 22.0 (95% CI 21.4, 22.5) and mean age at menarche, 11.5 years (95% CI 11.1, 11.8) years. In the 53 women in the Prospective Ovulation Cohort, breast experiences related to the menstrual cycle were analyzed within *all* 720 cycles, 97% of which were of normal lengths (21–36 days). We also analyzed them separately in the 694 of these cycles with documentable ovulatory status (Fig 1).

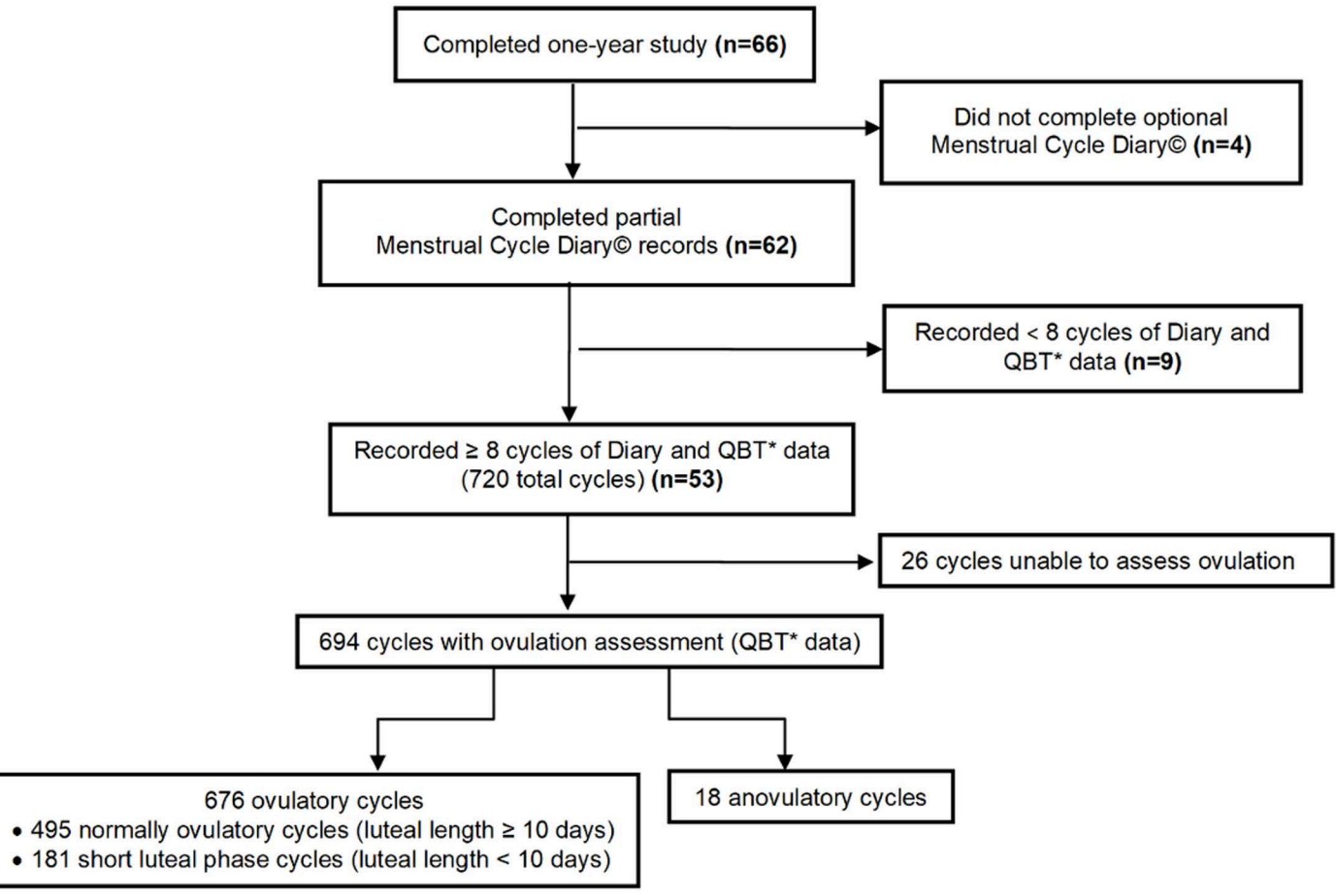

**Fig 1. Participant flow in the Prospective Ovulation Cohort (POC).** Flow of participants from the original POC to the 53 whose Diary data were analyzed for this breast tenderness and swelling study.

By QBT© analysis, 495 cycles were normally ovulatory (71%), 181 had short LL (<10 days, 26%), and 18 (3%) were anovulatory (SOD 199, 29%). Mean cycle lengths in all normally ovulatory cycles (495) and all anovulatory cycles (18) were also not different: 28.5 (95% 28.2, 28.7) and 27.8 (95% CI 25.8, 29.8), respectively. Of 53 participants, 47 women recorded *both* normally ovulatory and short luteal and anovulatory (SOD) cycles.

## Breast tenderness parameters

The mean intensity of breast tenderness in 53 women across all 720 cycles is shown in Fig 2 by quartiles of frequency distribution. The median breast tenderness was 1.4 (range, 0.0, 3.0) intensity in cycles of 28.1 days (95% CI 27.5, 28.8) long.

As shown in Table 1. no baseline data were different by median split of breast tenderness. All breast tenderness occurred in the luteal or premenstrual phase.

Breast tenderness parameters for all normally ovulatory versus SOD cycles showed that mean breast tenderness intensity and duration were significantly greater in normally ovulatory cycles (Table 2). The median Breast Tenderness Score was 6.0 (range 1.0–14.0) in the 495 normally ovulatory cycles versus 3.0 (0.0–11.0) for the 199 cycles with SOD (*P* =.005).

## Breast size changes

Changes in breast size from "usual" in the 711 cycles with this record were 4.0 (range 0.0–5.0). Breast size changes from usual were present for a mean of 4.0 (range 0.0–27.0) days/cycle and all occurred only in the luteal phase or premenstrually. Both increase in usual in breast size and its duration were significantly greater in normally ovulatory cycles (491) compared to SOD cycles (199). Breast swelling was scored (4.0 [2.0–4.0] vs. 4.0 [0–4.0]) (*P* =.034) and the days of breast enlargement 5.0 (1.0–11.0) vs. 3.0 (0–7.0) (*P* =.002) respectively (Table 2).

## Breast tenderness and breast size change parameters within-woman

We compared breast tenderness and size experiences related to ovulation status *within-woman* in 47 women (Table 3). Breast Tenderness Score for 420 normally ovulatory cycles was median 5.9 (0–33.0) for, and for the 199 SOD cycles was 5.7 (0–34.3). In normally ovulatory cycles the intensity of breast tenderness tended to be greater (1.24 vs 1.21, *P* =.089), and the duration of swelling (4.5 days vs 4.3) was significantly greater (*P* =.034).

## Breast tenderness and breast swelling temporal association in ovulatory cycles

In all ovulatory cycles aligned by QBT© ovulation day, we could integrate the cycle timing and both types of breast experiences by plotting mean breast tenderness and breast size changes per cycle day for the 676 cycles from 53 women (Fig 3). Tenderness and swelling were parallel in timing and positively correlated (r = 0.315, *P* =.001). The late luteal phase showed maximal values for timing of both breast characteristics.

## Discussion

These comprehensive 1-year healthy community cohort data described breast tenderness and breast swelling increases in the days before flow across menstrual cycles in 53 healthy, initially normally ovulatory women with the greatest changes being within normally ovulatory cycles. Mild breast tenderness is common and lasts 2–5 days. These data confirm previous literature about premenstrual breast tenderness and swelling but do not confirm our hypothesis that SOD cycles would be more symptomatic. We also showed a positive, significant correlation between breast tenderness and swelling.

It is accepted that regular, month-apart menstrual cycles are ovulatory [22,23]. We have recently shown that this is often untrue [24]. It is proposed that premenstrual experiences including breast tenderness and fluid retention relate to ovulation; they are thus unlikely to be reported in anovulatory cycles [8]. These ideas explained improved PMS on

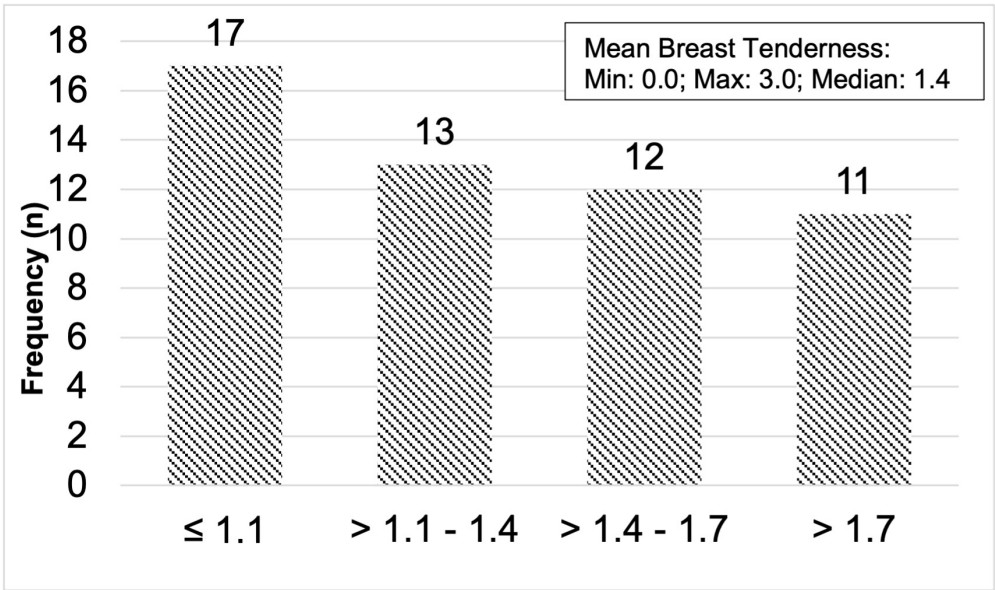

**Fig 2. Frequency distribution of breast tenderness.** Frequency distribution in quartiles of Mean Breast Tenderness experienced by 53 women (in all 720 recorded cycles) from the Prospective Ovulation Cohort. The number above each bar is the number of women in each quartile.

**Table 1. Demographics for all 53 POC Women.** Demographic, anthropomorphic, exercise, reproductive and menstrual health characteristics of 53 women (720 cycles) from the Prospective Ovulation Cohort reported by a median split of the intensity of breast tenderness.

| Characteristic | n = 53 | Intensity of Breast Tenderness | | P value |
| --- | --- | --- | --- | --- |
| | | Mild (<1.4) n = 30 | Moderate to Intense (≥ 1.4) n = 23 | |
| **†Mean and (95% confidence interval)** | | | | |
| Age (in years) | 33.9 (32.4-35) | 35.4 (33.5-37.3) | 32.6 (30.3 -35.0) | .066 |
| Height (in cm) | 162.5 (160.8-164.3) | 163.8 (161.1-166.5) | 161.3 (159.0-163.7) | .158 |
| Weight (kg) | 58.1 (56.2-60.0) | 59.5 (57.4-61.5) | 56.9 (53.8-59.9) | .165 |
| Body Mass Index (BMI) (kg/m$^2$) | 22.0 (21.4-22.6) | 22.2 (21.3-23.1) | 21.8 (21.0-22.6) | .449 |
| Age at Menarche (in years) | 11.5 (11.1-11.8) | 11.6 (11.2-12.0) | 11.3 (10.8-11.8) | .392 |
| Total Cycle # | 720 | 357 | 363 | – |
| Cycle (mean #/women) | 13.6 (12.8-14.4) | 14.3 (13.1-15.4) | 13.0 (11.9-14.1) | .093 |
| Cycle lengths (days) | 28.1 (27.5-28.8) | 27.9 (27.0-28.8) | 28.4 (27.4-29.3) | .444 |
| **‡Median and (range)** | | | | |
| Average duration of running (minutes/cycle) | 310.7 (0.0-1572.5) | 336.4 (0.0-1191.7) | 231.3 (0.0-1572.5) | .548 |
| Moderate-strenuous exercise^ (minutes/cycle) | 82.2 (0.0-854.2) | 59.1(0.0-477.1) | 104.4 (0.0-854.2) | .218 |
| | *n = 44* | *n = 20* | *n = 24* | |
| Lifetime # of months on CHC* | 42.0 (2.0-156.0) | 42.0 (2.0-143.0) | 42.0 (3.0-156.0) | .637 |
| | *n = 29* | *n = 13* | *n = 16* | |
| Lifetime months of pregnancy | 20.0 (2.0-40.0) | 20.0 (3.0-30.0) | 15.0 (2.0-40.0) | .619 |

*CHC means combined hormonal contraception.

^ *i.e.,* aerobics, swimming, skiing, fast walking, playing tennis, dancing.

*P* values are from either †ANOVA or ‡Mann-Whitney U Test.

A multiple linear regression analysis yielded no significant associations for relationships between breast tenderness and baseline variables.

**Table 2. Breast Tenderness and Size Change Parameters for all 53 POC Women.** Breast Tenderness and Breast Size Change Parameters between Normally Ovulatory and Ovulatory Disturbed Cycles by Quantitative Basal Temperature© from the Prospective Ovulation Cohort in all 53 Women.

| Characteristic Median (IQR³) | All 53 Women n = 694 cycles | Normally Ovulatory (Luteal Length ≥ 10 days) n = 495 cycles | Ovulatory Disturbances (Short Luteal phase & Anovulatory Cycles) n = 199 cycles | P Value* |
|---|---|---|---|---|
| Mean Intensity of Breast Tenderness (0–4 scale) | 1.0 (0.0 - 1.5) | 1.0 (1.0 - 1.5) | 1.0 (0.0 - 1.4) | .005 |
| Duration of Breast Tenderness (days/cycle) | 4.0 (0.0–9.0) | 4.0 (1.0–9.0) | 2.0 (0.0–8.0) | .001 |
| Breast Tenderness Score | 5.0 (0.0–12.0) | 6.0 (1.0–14.0) | 3.0 (0.0–11.0) | .001 |
| | n = 690 cycles | n = 491 cycles | n = 199 cycles | |
| Mean Change in Breast Size from Usual (per cycle) | 4.0 (0.0 - 4.0) | 4.0 (2.0–4.0) | 4.0 (0.0 - 4.0) | .034 |
| Duration of breast size change from usual (days/cycle) | 4.0 (0.0–10.0) | 5.0 (1.0–11.0) | 3.0 (0.0–7.0) | .002 |

*Mann-Whitney U Test ³IQR= Interquartile Range.

combined hormonal contraceptives (CHC) [8]. However, spontaneous ovulatory disturbances including anovulation (SOD) occur in up to one third of regular, normal-length menstrual cycles [16,23,25]. Population-based data in a previous study showed that cycles without ovulation by a cycle-timed serum progesterone level <9.5 nmol/L had both significantly lower progesterone and estradiol levels [23]. Although progesterone levels were 84% lower, estradiol was decreased by only 20% [23]. Based on the results of *our* study, it is possible that even this minimal decrease in estradiol was sufficient to result in a perceivable decrease in breast tenderness and swelling.

Few studies have previously sought to characterize menstrual cycle breast tenderness and size changes related to known ovulatory status [10–12] and no published study to date, has compared these in normally ovulatory vs. short luteal and anovulatory cycles *within* premenopausal women. In a study by Laessle et al. (1990) [11] data from prospective daily diary logs were used to analyze breast tenderness over a single cycle in 30 volunteer women (mean age 24) with confirmed ovulation by serum progesterone and known luteal phase lengths [11]. Our data confirm their observation of an increased breast tenderness later in the luteal phase [11]. They did not assess changes in breast size.

**Table 3. Within-woman Comparison of Breast Tenderness and Breast Size Change.** Within-woman comparison of Breast Tenderness Intensity, Duration, Breast Tenderness Score, and Mean Change in Breast Size (from usual = 3) and Number of Days of Breast Size Changes across all cycles in 47 women from the Prospective Ovulation Cohort who experienced both normally ovulatory cycles and subclinical ovulatory disturbances (anovulatory and cycles with luteal phase length <10 days).

| Characteristic Median (range) | Normally Ovulatory Cycles (LL ≥ 10) n = 420 | Subclinical Ovulatory Disturbances (LL < 10) n = 199 | Mean Positive Ranks | P Value[a] |
|---|---|---|---|---|
| Intensity of Breast Tenderness (0–4 scale) | 1.2 (0.0-2.3) | 1.21 (0.0-2.3) | 21.81 | .089 |
| Duration of Breast Tenderness (days/cycle) | 3.9 (0.0-16.0) | 4.3 (0.0-15.5) | 21.32 | .266 |
| Breast Tenderness Score | 5.9 (0.0-33.0) | 5.7 (0.0-34.3) | 20.29 | .268 |
| | n = 416 | n = 199 | | |
| Mean Change in Breast Size from Usual (per cycle) | 4.0 (0.0-4.4) | 4.0 (0.0-4.6) | – | .124 |
| Duration of breast size change from usual (days/cycle) | 4.5 (0.0-20.6) | 4.3 (0.0-19.0) | – | **.034** |

[a]Wilcoxon Signed Rank Test was used for these within-woman analyses.

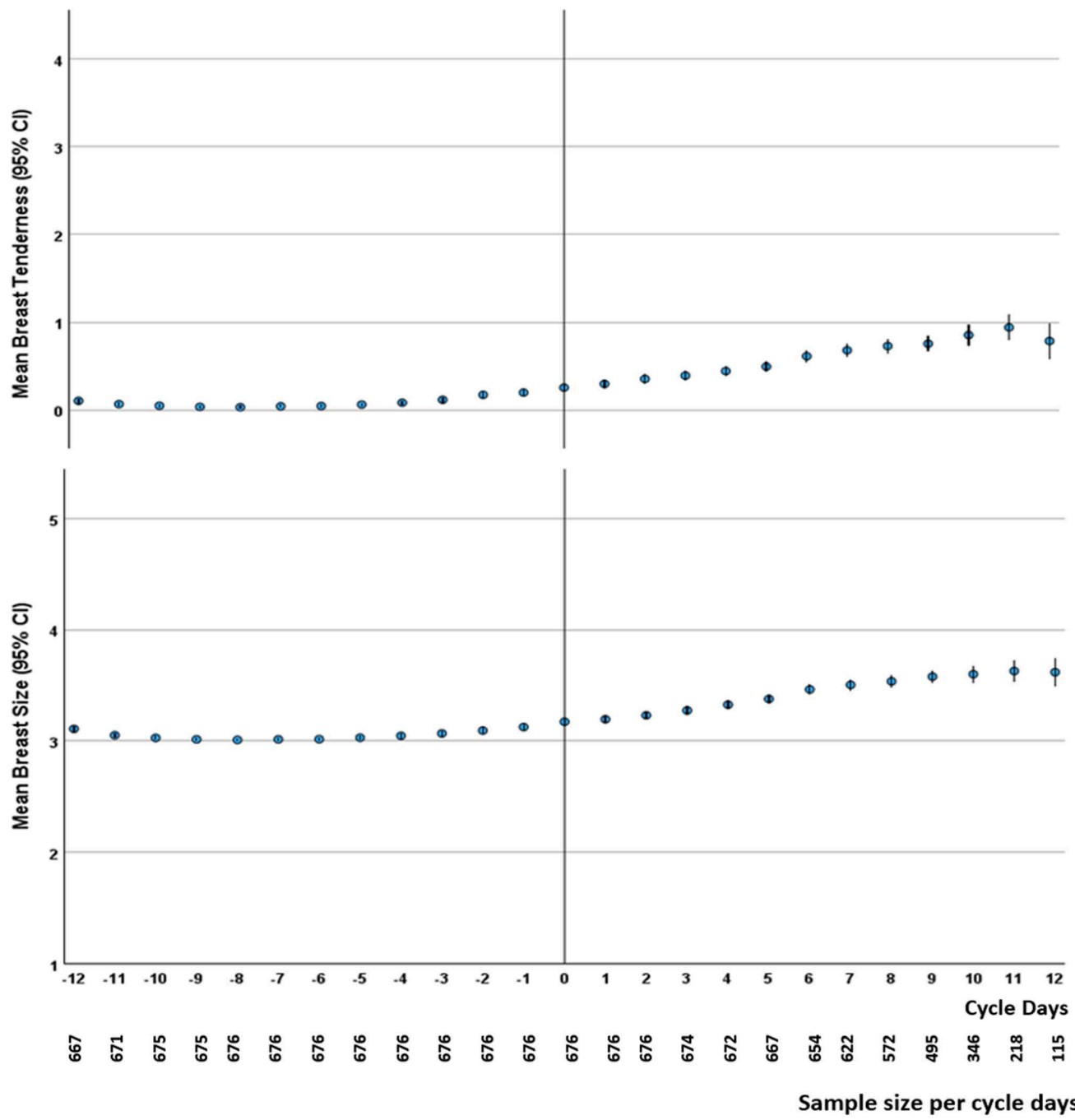

**Fig 3. Temporal pattern of mean breast tenderness and breast size change related to the day of ovulation.** Mean Breast Tenderness and Change from Usual in Breast Size (with 3 = usual) (95% CI) for 53 women (676 cycles) from the Prospective Ovulation Cohort are reported around the temperature shift day (ovulation = day '0') by Quantitative Basal Temperature© analysis.

Our centre, while evaluating women aged in their 40s with regular cycles also experiencing night sweats (clinically in perimenopause), found breast tenderness increased premenstrually in both anovulatory and ovulatory cycles [26]. In ovulatory cycles, alone, breast tenderness was also found to peak during the window of ovulation [26]. This temporal pattern was not present in our premenopausal cohort.

It is thought that high estradiol levels, combined with lower-than-normal levels of progesterone, are associated with increased breast tenderness [7,23,26]. Population data showed a higher estradiol/progesterone ratio that could result in proliferation of breast epithelial cells [27]. By contrast, the entire data set of our study documented that breast tenderness intensity and duration were *increased* in normally ovulatory cycles compared to cycles with disturbed ovulation.

Given the few (n =18) anovulatory cycles in our dataset, we had insufficient power to assess breast tenderness and size changes with anovulation. Further research is needed to relate estradiol and progesterone levels with breast tenderness and size changes.

Our secondary objective was to characterize the timing and variation of *normal* breast size changes associated with the menstrual cycle. There are studies examining breast changes in clinical populations and in women with 'clinical cyclical mastalgia' [1,28], but these experiences cannot be generalized to represent typical breast changes experienced over a normal menstrual cycle [10].

We also examined changes from each woman's perceived usual in breast size as well its relationship with breast tenderness [3,9]. Morphological changes in breast tissue during the luteal phase, including stromal edema, inflammatory cell infiltrates, and increased mitotic figure counts described by Ramakrishnan *et al.* [29], likely explain the significant association we found for breast swelling with breast tenderness. The extended exposure to luteal phase progesterone levels driving these morphological changes may also explain the increased days of breast swelling in normally ovulatory vs. SOD cycles.

Our study is limited as a secondary analysis of previously collected data. Direct measurements of estradiol and progesterone would have been ideal [7]. Funding insufficiency and study burden prohibited this. We also had few anovulatory cycles which likely limited our ability to show a within-woman difference in breast tenderness results.

Our study's strengths are prospectively reported data, minimizing recall bias, and characterizing diverse daily cycle experiences [1,2,14]. Its approximately 1-year duration also increases its validity versus many existing shorter studies [9,11,12]. Another strength is that all women on two consecutive cycles were initially normally cycling and ovulatory based on QBT© prior to enrollment. These breast experiences may be expected for healthy, initially ovulatory, premenopausal women.

## Conclusion

This one-year prospective observational cohort study in 53 healthy, normally cycling, and ovulatory women evaluated breast experiences recorded daily in 720 cycles. Results showed that mild breast tenderness and swelling occurred before flow in cycles with normal ovulation; these were totally absent during the follicular phase. Further research is needed on premenopausal normal breast experiences related to ovarian hormone levels.

## Acknowledgments

We thank the women who conscientiously kept daily records in the Prospective Ovulation Cohort. We also appreciate all those whose efforts contributed to the digitization and data management, including Yvette M. Vigna for review and documentation of the QBT© data. We thank Dr. Sewon Bann (working with co-author, Azita Goshtasebi) who determined that MCD© records in the POC were best evaluated in the 53 women with ≥8 menstrual cycle records/person (mean = 13).

## Author contributions

**Conceptualization:** Jerilynn C. Prior.

**Data curation:** Sonia Shirin, Azita Goshtasebi.

**Formal analysis:** Mary Wood, Sonia Shirin, Jerilynn C. Prior.

**Investigation:** Mary Wood, Jerilynn C. Prior.

**Methodology:** Sonia Shirin, Azita Goshtasebi, Jerilynn C. Prior.

**Resources:** Jerilynn C. Prior.

**Software:** Sonia Shirin.

**Supervision:** Jerilynn C. Prior.

**Validation:** Jerilynn C. Prior.

**Visualization:** Sonia Shirin.

**Writing – original draft:** Mary Wood.

**Writing – review & editing:** Mary Wood, Sonia Shirin, Azita Goshtasebi, Jerilynn C. Prior.

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
