## [Decision Letter · Decision Letter 0]

29 Dec 2024

PONE-D-24-53242Breast Experiences related to Menstrual Cycles and Ovulation in Healthy Premenopausal Women: the 1-year “Prospective Ovulation Cohort”PLOS ONE

Dear Dr. Prior,

Thank you for submitting your manuscript to PLOS ONE. After careful consideration, we feel that it has merit but does not fully meet PLOS ONE’s publication criteria as it currently stands. Therefore, we invite you to submit a revised version of the manuscript that addresses the points raised during the review process.

We look forward to receiving your revised manuscript.

Kind regards,

Patrick Ifeanyi Okonta, MBBCh, MPH, FWACS, FMCOG, MD, DRH

Academic Editor

PLOS ONE

Journal Requirements:

3. In this instance it seems there may be acceptable restrictions in place that prevent the public sharing of your minimal data. However, in line with our goal of ensuring long-term data availability to all interested researchers, PLOS’ Data Policy states that authors cannot be the sole named individuals responsible for ensuring data access (http://journals.plos.org/plosone/s/data-availability#loc-acceptable-data-sharing-methods).

Additional Editor Comments (if provided):

Dear Authors,

Kindly address the issues raised by the reviewers as stated in the added reviewer comment note

Reviewers' comments:

Reviewer's Responses to Questions

**Comments to the Author**

1. Is the manuscript technically sound, and do the data support the conclusions?

Reviewer #1: Yes

Reviewer #2: Partly

2. Has the statistical analysis been performed appropriately and rigorously? 

Reviewer #1: Yes

Reviewer #2: No

3. Have the authors made all data underlying the findings in their manuscript fully available?

Reviewer #1: Yes

Reviewer #2: No

4. Is the manuscript presented in an intelligible fashion and written in standard English?

Reviewer #1: Yes

Reviewer #2: Yes

5. Review Comments to the Author

Reviewer #1: The authors has looked into an aspect of medicine that has been assumed, their study is therefore opening new grounds for other researchers to look into.

But 40 years before the data was analyzed may be a drawback, I recommend a recent study can be done to evaluate the same breast changes.

The statistical analysis is appropriate.

The language was standard English and written in an intelligible fashion.

The conclusion is supported by the data.

Reviewer #2: The comments made have been uploaded to this page, in a pdf file. It will give an appropriate insight with respect to the the appropriate title, use of secondary data analysis, sampling techniques utilized in initial study and many more corrections.

6. PLOS authors have the option to publish the peer review history of their article (what does this mean? ). If published, this will include your full peer review and any attached files.

**Do you want your identity to be public for this peer review?** For information about this choice, including consent withdrawal, please see our Privacy Policy .

Reviewer #1: **Yes: ** Afeyodion Akhator

Reviewer #2: **Yes: ** OSAJI NICOLE EVELYN CHIUDE

---

## [Author Response · Author response to Decision Letter 0]

24 Feb 2025

February 11, 2025

Dear Dr. Patrick Ifeanyi Okonta,

1. Please ensure that your manuscript meets PLOS ONE's style requirements, including those for file naming. The PLOS ONE style templates can be found at https://journals.plos.org/plosone/s/file?id=wjVg/PLOSOne_formatting_sample_main_body.pdf and https://journals.plos.org/plosone/s/file?id=ba62/PLOSOne_formatting_sample_title_authors_affiliations.pdf<.https://track.editorialmanager.com/CL0/https:%2F%2Fjournals.plos.org%2Fplosone%2Fs%2Ffile%3Fid=ba62%2FPLOSOne_formatting_sample_title_authors_affiliations.pdf/1/010f0194149fe789-71bb2081-c775-402b-af26-b5b01e31d061-000000/Q8tPGFu9EXaoIM_gHIxp6uE6Q9jJ-zPEhRCjHu34xnA=190>

We have followed these instructions and believe that this manuscript meets PLOS ONE’s style requirements.

We fully support PLOS ONE’s efforts to make scientific primary data more easily available and potentially subject to re-analysis.

However, at the time of recruitment in the mid-1980s, the informed consent document included the scientific analysis and later use of these data. But it did not include that the primary data would be shared. This is a major ethical limitation.

2. In this instance it seems there may be acceptable restrictions in place that prevent the public sharing of your minimal data. However, in line with our goal of ensuring long-term data availability to all interested researchers, PLOS’ Data Policy states that authors cannot be the sole named individuals responsible for ensuring data access (http://journals.plos.org/plosone/s/data-availability#loc-acceptable-data-sharing-methods).

We agree that institutional management is more reliable. As the senior author, I am now almost 82 years old!

We are in the process of depositing our de-identified data for this analysis into a University of British Columbia “dataverse” instrument (Borealis) that will be accessible, if appropriate, through non-author university employees.

Additional Editor Comments (if provided):

We respectfully request that the revision of this manuscript proceed without interruption while these data availability issues are being resolved.

Thank you for your consideration of this request.

Detailed responses to reviewers including identification of manuscript changes

Reviewer 1

The authors has looked into an aspect of medicine that has been assumed, their study is therefore opening new grounds for other researchers to look into. But 40 years before the data was analyzed may be a drawback, I recommend a recent study can be done to evaluate the same breast changes.

The statistical analysis is appropriate. The language was standard English and written in an intelligible fashion. The conclusion is supported by the data.

Thank you for your positive assessments of the statistics, language and conclusion.

Unfortunately, at this time funding is not available to collect new data and repeat the initial study which included MCD© and QBT© data over the course of a year for 53 women. We do not believe that is necessary. There is no reason to think that contemporary data collection with repeated analysis would alter our findings. It is unlikely that spontaneously recorded breast tenderness and swelling experiences in relation to ovulatory status would have changed over a 40-year period. When looked at that way, we trust you will agree.

Reviewer 2

1.Cohort study done using previous secondary data is called a retrospective cohort study or historical cohort study. Secondary data cannot be utilized to carry out a prospective cohort study.

Thank you for your careful and comprehensive review.

You are quite correct that this is a secondary analysis. However, it is an analysis of prospectively collected data and thus still considered to have a prospective cohort design. The Menstrual Cycle Diary© (MCD©) data were collected prospectively during the initial study (Prior, 1990, New England J Med). In that investigation we used the MCD© for cycle length timing and as a form for recording first morning temperatures used in the Quantitative Basal Temperature© [QBT©] analysis.

However, most MCD data items were not reported until we determined the protocol for secondary analysis of these diary data in the Prospective Ovulation Cohort study analyzing and reporting the MCD©-recorded experiences of the 53 women with at least 8 consecutive cycles (mean 13.6). That published open-access protocol is accessible here http://hdl.handle.net/2429/80682.

This is a prospective study because the participants were followed prospectively (over time into the future) and as stated by Klebanoff & Snowden (2018). It is the timing of data collection with respect to when the follow up/observed outcome takes place that makes a study prospective vs. retrospective, not the timing of the analysis, nor the year when the data were collected.

Klebanoff MA, Snowden JM. Historical (retrospective) cohort studies and other epidemiologic study designs in perinatal research. American Journal of Obstetrics & Gynecology. 2018;219(5):447-50.

ABSTRACT

2. Title should contain “A secondary data analysis “

Thank you for that suggestion. We have added it (Line 4).

3. A standard definition of the population used which falls into the pre- menopausal age group bracket. Your population was 20-40 years are they pre-menopausal? Insert your reference.

The participants included in this study were deemed premenopausal. They had predictable normal length (21-36 day) menstrual cycles with two consecutive normally ovulatory cycles prior to study enrolment. Normal ovulatory status was proven with QBT© determined luteal phase lengths of 10 or more days. Based on the STRAW+10 criteria (Harlow et al., 2012), they would only be considered perimenopausal if they had persistent variability in cycle lengths by 7 days or more in consecutive cycles.

Harlow SD, Gass M, Hall JE, Lobo R, Maki P, Rebar RW, et al. Executive summary of the Stages of Reproductive Aging Workshop + 10: addressing the unfinished agenda of staging reproductive aging. Menopause. 2012;19(4):387-95.

4. Title is vague should have indicated the specific breast experiences which will be the dependent variable or outcome. As prospective cohort studies must have a timeline for follow up, which indicates the time to develop the outcome (and exemplified as per person per year) not any arbitrary time frame.

Thank you for that suggestion. We have added “breast tenderness and swelling” to the title (Line 3).

5. For range in the statement (Line 26), the comma should be changed to hyphen “the median breast tenderness was 1.4 (on a 0-4 scale, range 0.0, 3.0)”.

We generally prefer a comma between ranges to ensure that a hyphen is not interpreted as a negative number. However, we have accepted your suggestion. See (Line 28)

6. For CI in the statement ( line 27 ) “ in cycles with a mean length of 28.1 days (95% CI 27.5, 28.8)”. The comma should be changed to hyphen.

We have again accepted your suggestion. See (Line 29)

7. Explain how the within woman finding of no statistical significance in breast tenderness impacts on your primary objective of your study and its conclusion. Your conclusion in the abstract should be in line with the objectives, this enables relay a snapshot summary.

We respectfully disagree with the suggestion to describe the primary outcome first.

Given that there is reason to suspect more accuracy of a self-reported experience within- than between women, that the within-woman outcome was our primary objective, and that our within-woman and between women results differed, we believe that our within-woman findings are important to report in the abstract. We had only 18 anovulatory cycles in this database given that all women were initially normally cycling and ovulatory. These results highlight the need for future research with more anovulatory cycles for direct comparison to further evaluate our primary objective.

8. The study design utilized is limited when temporality needs to be established hence conflicting with your statement in line 52-54 which states “Both women and clinicians need to know what is expected or ‘normal’ for menstrual cycle related breast swelling and tenderness. We also need to know if breast changes across a cycle reflect ovulatory status”, cannot be investigated extensively with regard to casualty.

You are right that we cannot assess causality. We are only attempting to describe breast tenderness and swelling experiences as well as when within the menstrual cycle the breast experience changes occurred (temporality).

Our study prospectively followed 53 previously healthy and normally cycling women with self-reported breast experiences (tenderness and size changes) over at least 8 consecutive cycles (mean 13). We were able to show the general trends and characterization of breast changes experienced across cycles that were proven to be ovulatory based on QBT© data, as well as the relationship and cycle phase timing of breast tenderness and swelling throughout all ovulatory cycles.

Given the extensive screening, and the healthy nature of the participants, these findings can be extrapolated to similar populations as a reflection of expected menstrual cycle experience changes.

INTRODUCTION

A. This statement should be referenced “However, most menstrual studies do not assess ovulatory status or use non-valid methods”.

We agree that this statement should be referenced. We have added two references, (Lines 51-52)

B. The normal menstrual cycle length is 21-35 days. Reference this statement “To enrol, women needed two consecutive normal-length (21-36 day)

You are right to call this out.

In the late 1970s when planning was underway for data collection, the upper length of the normal menstrual cycle was considered 36 by experts like Dr. Abraham. We’ve referenced a review of his (Line 73).

C. Line 6. How did you account for differential pain perception- minimal pain documented by a respondent may be maximal to respondent (confounding/ propensity scoring)

Pain scales are entirely subjective, and as you correctly identified, respondents have different pain thresholds and tolerance levels. We were assessing breast tenderness that we expected to be and was usually mild (as the median was 1.4 on a 0-4 scale shows).

We simply scored breast tenderness with the same scale as we used for other things like frustration or sleep problems. Our MCD© data were to evaluate trends and variations in reported experiences throughout an individual’s cycle to assess patterns with regards to ovulatory status.

D. Line 108: Delete this ( ) from the range and insert a comma, in the statement “as mean (95% confidence interval) or median (range) across two groups”. It is a distinct statistical term, a measure of dispersion not a measure of central tendency.

We have followed your suggestion—see (Lines 112-113)

E. Line 110: correct analysis in the statement “Based on data distribution (normally vs. non-normally distributed), statistical analyses” analysis is singular while analyses is plural.

Thank you. We have changed ‘analyses’ to ‘analysis’ (Line 114)

G. Line 112-113: You used Man Whitney U test {which is utilized for non- parametric data }to compare the mean (a parametric data ).

In summarizing, I guess our statistical analysis plan became unclear.

We have indicated that we used a non-parametric test for data (such as the ordinal and not interval data in which breast tenderness was scored) that were not normally distributed.

H. Line 115-118: What objective are you analyzing using Wilcoxon Signed Rank Test, when you compared the constructed composite “breast tenderness score” with the other parameters in both groups?

Thank you for this question. In Table 3 we should have made it clear that our non-parametric but paired analysis of breast experience parameters used this test for within-woman analysis. We’ve added a footnote to Table 3 to indicate the use of the Wilcoxon Signed Rank Test for these data.

I. Line 122-123: What objective did breast tenderness and size correlations answer. Always do the statistical analysis which are pertinent to your study objectives.

Thank you for the important reminder to make sure that data analysis responds to stated study objectives.

One of our secondary objectives was to characterize the timing and variation of normal breast size changes associated with the menstrual cycle (Lines 301-302). As a component of this analysis, we examined the timing and pattern of both breast tenderness and change in size across a normally ovulatory cycle. These two breast experiences were found to be positively correlated, both increasing towards the end of the luteal phase prior to the onset of menses.

Knowing that increased breast tenderness and increased breast size or swelling, which can contribute to perceived breast tenderness, is a normal premenstrual experience across an ovulatory cycle is valuable information to both clinicians and premenopausal women in the context of menstrual health evaluation.

ETHICAL APPROVAL

A. Line 133-134: This statement is not necessary “These data were de-identified so that the identities of the participants were not known to the researchers analyzing the data”.

We have taken this statement out (see Line 142) but referred to de-identified data elsewhere as our ethics confidentiality guideline require.

B. This statement should be under the section on ethical approval “ They were not financially compensated for study participation”.

We have moved the place in the document in which this is mentioned, as you requested. (Lines 140-142)

C. Insert the appropriate flow diagram for the study.

We have inserted the Flow Diagram early in the results to orient the reader to what data we are analyzing (see Lines 168-171)

RESULTS

Documentation should chronologically reflect your objectives to reduce ambiguity.

Thank you for your perspective.

Before the objectives can be understood and data presented, we believe it is important to first introduce the participants in a trial, to show the flow of participants and data through the study, to show the distribution of the major factor of interest (breast tenderness) and to first provide complete baseline data stratified by the median breast tenderness.

We then show the primary parameters for breast tenderness and size changes for all women, examining

---

## [Editor Report · Decision Letter 1]

3 Mar 2025

Breast Tenderness and Swelling Experiences Related to Menstrual Cycles and Ovulation in Healthy Premenopausal Women: secondary analysis of the 1-year “Prospective Ovulation Cohort”

PONE-D-24-53242R1

Dear Dr. Prior,

We’re pleased to inform you that your manuscript has been judged scientifically suitable for publication and will be formally accepted for publication once it meets all outstanding technical requirements.

Kind regards,

Patrick Ifeanyi Okonta, MBBCh, MPH, FWACS, FMCOG, MD, DRH

Academic Editor

PLOS ONE
---

## [Editor Report · Acceptance letter]

PONE-D-24-53242R1

PLOS ONE

Dear Dr. Prior,

I'm pleased to inform you that your manuscript has been deemed suitable for publication in PLOS ONE. Congratulations! Your manuscript is now being handed over to our production team.

Kind regards,

on behalf of

Professor Patrick Ifeanyi Okonta

Academic Editor

PLOS ONE